**Short summary.** Wolverine denning habitat inferred using a snow threshold differed for three different spatial
representations of snow. These differences were based on the annual volume of snow and the elevation of the snow
line. While denning habitat was most influenced by winter meteorological conditions, our results show that studies
applying thresholds to environmental datasets should report uncertainties stemming from different spatial
resolutions and uncertainties introduced by the thresholds themselves.
# Interactions between thresholds and spatial discretizations of snow: insights
# from estimates of wolverine denning habitat in the Colorado Rocky
# Mountains
Justin M. Pflug[1,a,b], Yiwen Fang[2], Steven A. Margulis[2], Ben Livneh[1,3]
[1]Cooperative Institute for Research in Environmental Science (CIRES), University of Colorado,
Boulder, CO, 80309, USA
[2]Department of Civil and Environmental Engineering, University of California, Los Angeles, CA,
90095, USA
[3]Department of Civil, Environmental and Architectural Engineering, University of Colorado, Boulder,
CO, 80309, USA
[a]now at: Hydrological Sciences Laboratory, NASA Goddard Space Flight Center, Greenbelt, MD,
20771, USA
[b]now at: ESSIC, University of Maryland, College Park, College Park, MD, 20742, USA
*Correspondence to*: Justin M. Pflug (jpflug@umd.edu)
**Abstract.** Thresholds can be used to interpret environmental data in a way that is easily communicated and useful
for decision making purposes. However, thresholds are often developed for specific data products and time periods,
changing findings when the same threshold is applied to datasets or periods with different characteristics. Here, we
test the impact of different spatial discretizations of snow on annual estimates of wolverine denning opportunities in
the Colorado Rocky Mountains, defined using a snow water equivalent (SWE) threshold (0.20 m) and threshold date
(15 May) from previous habitat assessments. Annual potential wolverine denning area (PWDA) was thresholded
from a 36-year (1985 – 2020) snow reanalysis model with three different spatial discretizations: 1) 480 m grid cells
(D480), 2) 90 m grid cells (D90), and 3) 480 m grid cells with implicit representations of subgrid snow spatial
heterogeneity (S480). Relative to the D480 and S480 discretizations, D90 resolved shallower snow deposits on
slopes between 3050 and 3350 m elevation, decreasing PWDA by 10%, on average. In years with warmer and/or
drier winters, S480 discretizations with subgrid representations of snow heterogeneity increased PWDA, even within
grid cells where mean 15 May SWE was less than the SWE threshold. These simulations increased PWDA by
upwards of 30% in low snow years, as compared to the D480 and D90 simulations without subgrid snow
heterogeneity. Despite PWDA sensitivity to different snow spatial discretizations, PWDA was controlled more by
annual variations in winter precipitation and temperature. However, small changes to the SWE threshold ($\pm$ 0.07 m)
and threshold date ($\pm$ 2 weeks) also affected PWDA by as much as 82%. Across these threshold ranges, PWDA was
approximately 18% more sensitive to the SWE threshold than the threshold date. However, the sensitivity to the
threshold date was larger in years with late spring snowfall, when PWDA depended on whether modeled SWE was
thresholded before, during, or after spring snow accumulation. Our results demonstrate that snow thresholds are
useful but may not always provide a complete picture of the annual variability in snow-adapted wildlife denning
opportunities. Studies thresholding spatiotemporal datasets could be improved by including 1) information about the
fidelity of thresholds across multiple spatial discretizations, and 2) uncertainties related to ranges of realistic
thresholds.
## 1. Introduction
Generalizing environmental data using thresholds can present information in a way that is more easily understood,
communicated, and applied for decision-making purposes. Conceptually, thresholds are static constraints intended to
partition the areas, timing, and/or prevalence of data greater or less than some scientifically or managerially relevant
limit. In the field of snow science, thresholds are used to classify snow cover and snow absence from remotely-
sensed observations (Dozier, 1989; Hall and Riggs, 2007; Sankey et al., 2015), partition snow accumulation and
snowmelt seasons (Cayan, 1996; Hamlet et al., 2005; Mote et al., 2005; Serreze et al., 1999), and parameterize
modeled processes like snow-layer formation and merging (e.g., Clark et al., 2015; Liston and Elder, 2006;
Wigmosta et al., 2002), rain and snow precipitation partitions (Auer, 1974; Harder and Pomeroy, 2013), and snow
holding capacity on steep slopes (Bernhardt and Schulz, 2010). Thresholds are also used to identify drought
conditions in snow-dominated watersheds (Dierauer et al., 2019; Harpold et al., 2017; Heldmyer et al., 2023) , and
the associated "decision trigger" and "tipping point" thresholds that determine water use and allocation in regulated
basins (Herman and Giuliani, 2018; Kwadijk et al., 2010; Shih and ReVelle, 1995). However, despite widespread
use, thresholds are often developed for specific applications, and over short time intervals, decreasing the likelihood
that a threshold developed for one purpose could be applied in an identical manner to different periods of time, or to
environmental products with different characteristics (Härer et al., 2018; Jennings et al., 2018; Maher et al., 2012;
Pflug et al., 2019).
Here, we focus on snow thresholds that have been used increasingly over the past decade to identify regions with
conditions suitable for the survival of snow-adapted wildlife. Many studies use thresholds that focus on snow
characteristics like snow depth, snow cover, snow density, snow water equivalent (SWE), and snowmelt season
snow persistence, which can be important for denning, migration, and food-availability for species like wolverines
(*Gulo gulo*), polar bears (*Ursus maritimus*), and Dall sheep (*Ovis dalli dalli*) (Barsugli et al., 2020; Durner et al.,
2013; Liston et al., 2016; Mahoney et al., 2018; McKelvey et al., 2011; Sivy et al., 2018). However, relatively few
studies simulate snow at spatial resolutions that correspond to the features that drive snow habitat (e.g., Glass et al.,
2021; Liston et al., 2016; Mahoney et al., 2018). For instance, wolverines rely on snow drifts for maternal and natal
denning. These drifts often form alee of obstructions near the forest edge and in talus fields (e.g., Fig. 1, star). Yet,
few models simulate snow at den-scale spatial-resolutions (< 10 m), and represent the physical processes that
control the formation of dens, like wind-redistribution, preferential deposition, avalanching, and microtopographic
shading. This is particularly the case for species status assessments which often attempt to quantify wildlife habitat
at large regional extents where high-resolution snow simulations with complex physical processes would be
computationally prohibitive. Thresholds are therefore used to facilitate the relationship between a coarser-resolution
representation of snow, and the finer-scale feasibility of wildlife habitat. The validity of this approach is debated
(e.g., Araújo and Peterson, 2012; Barsugli et al., 2020; Boelman et al., 2019; Bokhorst et al., 2016; Copeland et al.,
2010; Magoun et al., 2017). For example, coarser-scale representations of snow may resolve the larger-scale
meteorological influences on habitat availability, but coarser-scale representations of snow likely overlook the
smaller-scale refugia that could continue to support habitat, even with future changes to climate.
This study builds on work from Barsugli et al. (2020), who used physically-based simulations to identify regions
that could support wolverine denning using a SWE threshold (0.20 m) on a static date (15 May) corresponding to the
tail end of the maternal denning period(Copeland et al., 2010; McKelvey et al., 2011; USFWS, 2018). This 0.20 m
SWE threshold was chosen based on 15 May SWE that corresponded to known wolverine denning sites from a 250
m snow simulation (Barsugli et al., 2020; Ray et al., 2017; USFWS, 2018). Barsugli et al. (2020) found that, relative
to previous studies that used ~10 km products (Laliberte and Ripple, 2004; McKelvey et al., 2011), snow
simulations at 250 m resolution were able to better resolve SWE persistence, and increased habitat, on shaded north-
facing slopes. 250 m simulations also increased the overall prevalence of snow that could support wolverine dens,
both in current and future climates, over Colorado and Montana Rocky Mountain domains.
Here, we extend the findings from Barsugli et al. (2020), testing the difference in wolverine denning support defined
using thresholds (0.20 m SWE on 15 May) and a historic snow reanalysis with different spatial discretizations (Fig.
1). These discretizations include: 1) discrete 480 m grid cells (D480), 2) discrete 90 m grid cells (D90), and 3) 480
m grid cells with implicit representations of subgrid SWE spatial heterogeneity (S480). These discretizations
straddle the 250 m resolution used by Barsugli et al. (2020) and include both discrete (D480 and D90) and implicit
(S480) representations of snow distribution. These reanalyses, which combine snow modeling and remotely-sensed
observations of snow cover (more in Sect. 2.2), also resolve snow volume and distribution in mountain terrain
significantly better than more common modeling approaches (Pflug et al., 2022; Yang et al., 2021). We focus on the
same Colorado Rocky Mountain domain used by Barsugli et al. (2020) over a longer period of 36 years, spanning
1985 to 2020. We address the following research questions: **1) how does the spatial discretization of snow**
**influence estimates of potential wolverine denning area (PWDA)? and 2) is the sensitivity of PWDA to**
**different snow spatial discretizations greater or smaller than the sensitivity to annual changes in winter**
**climatic conditions?** We also identify the spatial locations and causes of the greatest differences PWDA, and
evaluate sensitivities to small uncertainties in both SWE thresholds ($\pm$ 0.07 m) and threshold dates ($\pm$ 2 weeks).
More generally, this study highlights shortcomings, opportunities, and tradeoffs to thresholding spatial snow
products, and serves as a roadmap for future wildlife habitat assessments.

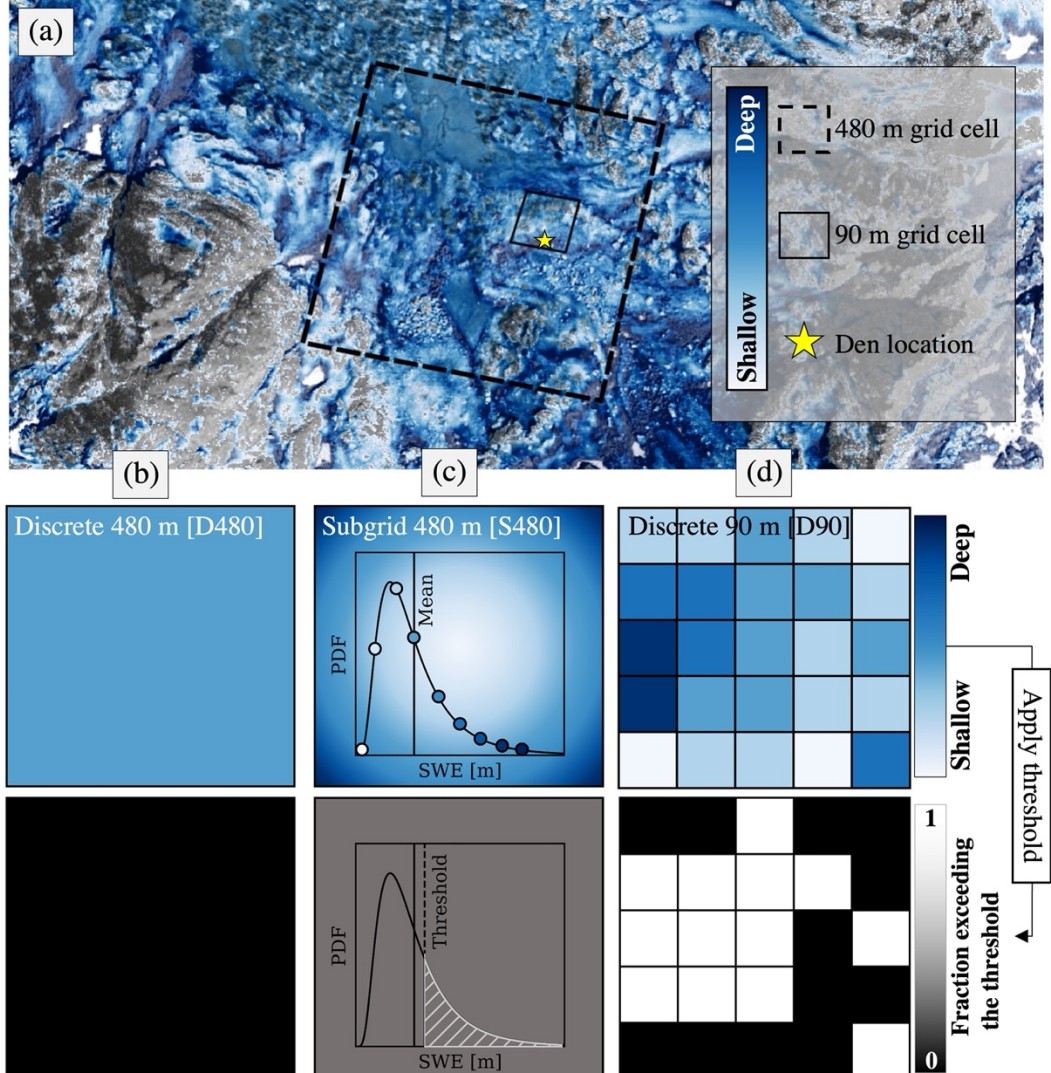


Figure 1. SWE spatial heterogeneity inferred from airborne lidar at 1 m resolution, compared to 480 and 90 m grid
cells, and a point (star) with a snow drift suitably deep for wolverine denning (a). SWE is simulated in this study
using three different spatial discretizations: 480 m discrete grid cells (D480, column b), 480 m grid cells with
subgrid SWE heterogeneity (S480, column c), and 90 m discrete grid cells (D90, column d).  The fraction of the area
that could support wolverine denning is estimated for each discretization using a 0.20 m SWE threshold on 15 May.
The fraction of the area exceeding the SWE threshold is binary (fully greater than or less than the threshold) for
discrete grid cells (b and d), while the area exceeding the SWE threshold for the S480 discretization (c) is defined by
the fraction of the grid cell SWE distribution exceeding the threshold (white hatching)
**2. Domain and Data**
**2.1. Domain**
We focused this work over Rocky Mountain National Park in Colorado state (Fig. 2). This domain is home to
several snow-adapted wildlife species, and has been included in wolverine habitat assessments (Barsugli et al., 2020;
McKelvey et al., 2011; USFWS, 2018). Barsugli et al. (2020) estimated most of the terrain supportive of wolverine
habitat in this region to be between 2700 and 3600 m of elevation. Although this area does not currently support a
reproductive population of wolverines, this region is of potential interest for wolverine reintroduction. More
information about wolverine habitat can be found in the U.S. Fish and Wildlife Service species status assessment
(USFWS, 2018).
The Rocky Mountain National Park domain contained several snow observations (Fig. 2). These observations
include 28 snow telemetry (SNOTEL) stations, deployed and managed by the National Resources and Conservation
Service. These stations use snow pillows to measure the weight of snowpack and resulting SWE. A distributed lidar
observation of snow depth in southernmost portion of the domain was also collected by the National Center for
Airborne Laser Mapping in May 2010. These observations were used to assess the accuracy of the SWE reanalysis
discussed in Sect. 2.2.

## 2.2. SWE Reanalyses

SWE was calculated over the Rocky Mountain domain (Figure 2) from a popular satellite-era (water years 1985 –
2020) probabilistic snow reanalysis (Margulis et al., 2019, 2016, 2015) performed at 3 arcseconds (~90 m) and 16
arcseconds (~480 m). This reanalysis was generated at each individual grid cell using an ensemble of simulations
forced by the Modern-Era Retrospective analysis for Research and Applications, Version 2 (MERRA-2; Gelaro et
al., 2017), and simulated using the simplified Simple Biosphere Model, Version 3 (Xue et al., 1991) coupled with
the Liston (2004) snow depletion curve. The forcing dataset was downscaled to the simulation grid (Girotto et al.,
2014; Margulis et al., 2015) before running the land surface model. Model ensemble members were provided
different 1) precipitation multipliers (influencing total snow mass), 2) snow albedo decay functions (influencing the
rate of snow ablation), and 3) parameterizations of subgrid snow spatial variability (influencing subgrid snow cover
during snowmelt), among other parameters. The reanalysis then reweighted the ensemble members to most-heavily
favor those that matched the snowmelt season evolution of fractional snow covered area from 30 m Landsat
observations. We expect uncertainties and errors in the snow reanalysis owing to both errors in meteorological
forcing data (e.g., Daloz et al., 2020; Liu and Margulis, 2019) and errors with the snow model (e.g., Feng et al.,
2008; Xiao et al., 2021)However, the ensemble approach used by this reanalysis adjusted modeled snow
accumulation and depletion to track remote sensing observations of snow cover depletion, which has shown the
capability to bias-correct SWE and implicitly account for difficult-to-simulate processes like precipitation lapse
rates, wind-loading/scour, avalanching, and forest-snow processes (e.g., Pflug et al., 2022; Yang et al., 2021).
Relative to SNOTEL observations, which are not used by the snow reanalysis, the reanalysis exhibited a SWE
coefficient of correlation of 0.82  between 1985 and 2020 in the Rocky Mountain domain (Fig. S1). On average, the
reanalysis was biased low relative to the snow pillow observations by approximately 23%. However, this could be
attributed to the location of SNOTEL observations in forested clearings (Fig. 2a) which typically have SWE deeper
than the terrain covered by the 480 and 90 m pixels(e.g., Livneh et al., 2014; Pflug et al., 2022).While the snow
reanalysis used in this study is ultimately a model product and subject to a number of modeling uncertainties, the
SWE simulated by the 90 m and 480 m discretizations agreed closely with each other and with ground observations.
Therefore, spatial differences in 15 May SWE, and the resulting distribution of snow that exceeded the SWE
threshold (e.g., Fig. 1) was attributable to differences in the interactions between the static SWE threshold and
different spatial discretizations of snow.
For the 480 m grid cells with subgrid snow variability (Fig. 1c, S480), the heterogeneity of SWE was estimated
using a method developed by Liston (2004). This method assumes that the subgrid heterogeneity of SWE
accumulation is lognormally distributed, and is dictated by a time-constant coefficient of variation (CoV),

$$CoV = \frac{\sigma}{\mu},$$

*(1)*

where $\mu$ is the grid cell mean SWE and $\sigma$ is the standard deviation of the SWE within that grid cell. The CoV of
subgrid SWE accumulation (Fig. 2b and 2c) was determined for each 480 m grid cell using the most common
pattern of SWE accumulation from the overlapping 90 m reanalysis grid cells (Fig. 1d) between 1985 and 2020
(detailed further in Text S1). In Sect. 3.1, we discuss how CoV was used to estimate the temporal evolution of
subgrid SWE heterogeneity.

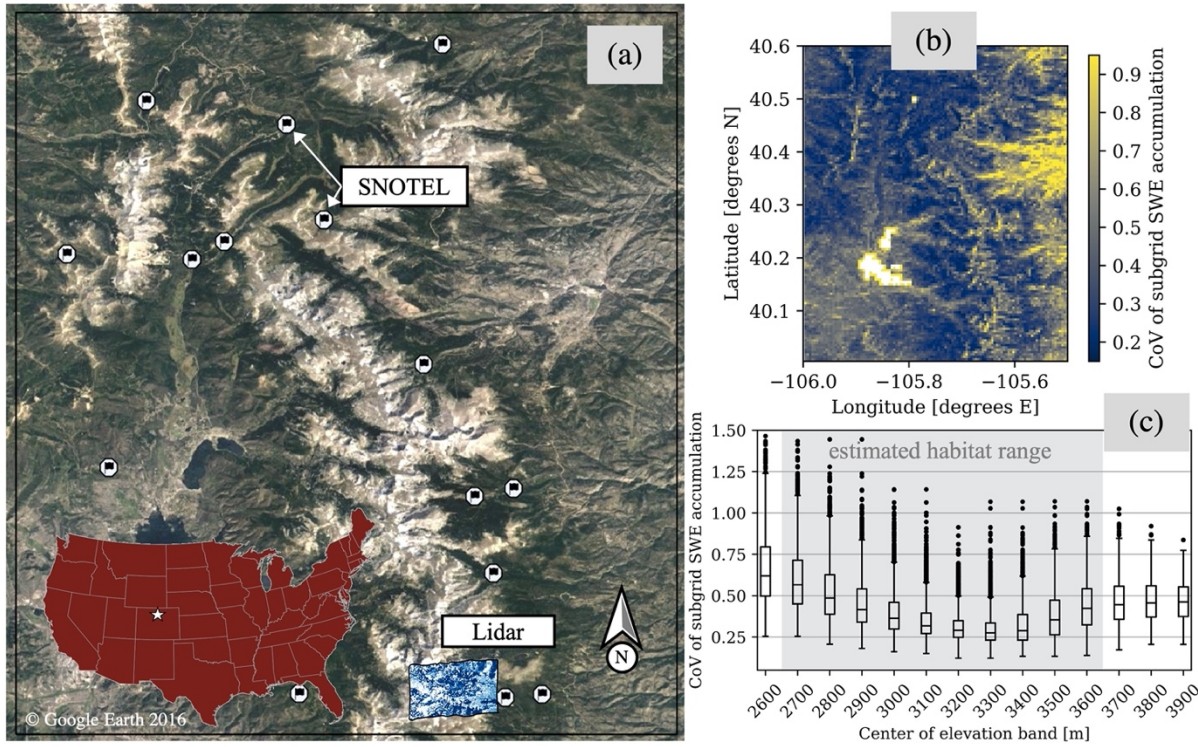


Figure 2. Rocky Mountain National Park study domain. The location of SNOTEL observations and lidar snow depth
observations are superimposed in the terrain map (a). The 480 m coefficient of variation of subgrid SWE
accumulation is shown both spatially (b) and across 100 m elevation bands (c).
**3. Methods**
The methods evaluate the impacts of snow spatial discretizations and winter climatic conditions on assessments of
total area suitable for denning wolverines. We investigated three different spatial discretizations; two discretizations
using more common discrete representations of snow, and one with an implicit representation of subgrid snow
heterogeneity (see Sect. 3.1). For each,  potential wolverine denning area (PWDA) was calculated using a static
SWE threshold (0.20 m) on a static spring date (15 May) (Sect. 3.2). Finally, we partitioned years with winter
precipitation magnitude and precipitation phase climate categories (wet, dry, cold, and warm) (see Sect. 3.3). These
categories were used to examine whether winter climatic conditions or model representations of snow spatial
distribution most-influenced estimates of PWDA.
**3.1. Subgrid SWE evolution**
The temporal evolution of subgrid SWE heterogeneity was estimated for 480 m grid cells (Fig. 1, S480) using
methods developed by Liston (2004) (Fig. 3). Provided the reanalysis grid cell mean SWE ($\mu$) from a D480grid cell
(Fig. 1b), and a CoV of subgrid SWE accumulation (Fig. 2b), the probability distribution of subgrid SWE for that
grid cell ($f(SWE)$) was calculated using a lognormal distribution,

$$f(SWE) = \left(\frac{1}{SWE\zeta\sqrt{2\pi}}\right)exp\left[-\frac{1}{2}\left[\frac{\ln(SWE)-\lambda}{\zeta}\right]^2\right],$$

*( 2 )*

$$\lambda = \ln(\mu) - \frac{1}{2}\zeta^2,$$

*( 3 )*

$$\zeta^2 = \ln(1 + CoV^2).$$

$$(4)$$

Figure 3b demonstrates the subgrid distribution of SWE in two winter periods ($t_a^1$ and $t_a^2$) assuming the mean SWE evolution from Fig. 3a, a CoV of 0.50, and Eq. 2 – 4.

In the snowmelt season, the Liston (2004) methodology assumes spatially-uniform snowmelt, causing snow disappearance first in locations with thinner SWE, and last in locations with deeper SWE. This can be conceptualized as taking the subgrid distribution of snow at peak SWE (Fig. 3b, $t_a^a$), and adjusting it downwards by a constant amount to reflect spatially-uniform melt ($SWE_m$) (Fig. 3c). In doing so, snow only exists for portions of the gridcell where $f(SWE)$ at peak SWE was greater than $SWE_m$. Therefore, the fractional snow-covered area (fSCA) of the grid cell could be calculated from the fraction of the distribution ($f(SWE)$) with SWE greater than $SWE_m$,

$$fSCA = \int_{SWE_m}^{\infty} f(SWE)dSWE.$$

$$(5)$$

Since $SWE_m$ can exceed the amount of SWE that exists in some locations at peak SWE timing, and since SWE cannot be less than 0 m (snow-absent), the change in gridcell mean SWE ($\mu$) throughout snowmelt will not necessarily equal $SWE_m$. Rather, $\mu$ throughout the snowmelt season can be calculated from the expected value of the melt-shifted distribution (Fig. 3c),

$$\mu = \int_{SWE_m}^{\infty} [SWE - SWE_m]f(SWE)dSWE.$$

$$(6)$$

In this study, we were provided $\mu$ from the reanalysis at each 480 m grid cell and daily timestep. Using the CoV calculated from the overlapping D90 data (Fig. 2b), and maximum annual $\mu$ at each grid cell, we calculated the SWE distribution (Eq. 2) for each grid cell at peak SWE timing. Then, using a Newton-Raphson solver, we solved the $SWE_m$ for each grid cell that caused $\mu$ from Eq. 6 to match D480 $\mu$ at each grid cell on 15 May.

The Liston (2004) subgrid SWE parameterization discussed above operates under several assumptions. Like many other studies (e.g., Donald et al., 1995; Helbig et al., 2021; Jonas et al., 2009), Eq. 2 assumes that the distribution of snow accumulation at scales finer than the grid cell resolution can be represented by a lognormal distribution. We tested this assumption by evaluating the distribution of 1 m lidar snow depth observations (Fig. 2a) that fell within 480 m grid cells. The Kolmogorov-Smirnov (KS) statistic, or maximum difference between cumulative distribution functions, was used to test how well different theoretical distributions (e.g., normal, lognormal, gamma, Rayleigh, chi, etc.) used by a variety of snow studies (e.g., He et al., 2019; Helbig et al., 2015; Mendoza et al., 2020; Pflug and Lundquist, 2020; Skaugen and Melvold, 2019) matched the lidar-observed snow depth distributions. The KS statistic for the lognormal distribution (Eq. 2) was $0.12 \pm 0.05$, and was significantly worse (greater than 0.22) when comparing the observed lidar distributions versus other common distributions, like normal and gamma distributions. While not perfect, these results showed that subgrid snow heterogeneity was approximated best by lognormal distributions. The Liston (2004) subgrid methodology also assumed that the CoV of subgrid SWE accumulation was constant, resulting in a linear increase in SWE variability (standard deviation) with mean SWE throughout the snow accumulation season (Fig. 3b). While we lacked validation data to test this, this assumption is the basis for other modeling approaches, which scale snow input using information from historic snow accumulation patterns (Liston, 2004; Luce et al., 1998; Pflug et al., 2021; Vögeli et al., 2016). Finally, although subgrid snowmelt is not spatially-uniform, melt-season snow heterogeneity is often modeled well by assuming uniform snowmelt. This is due to the outsized influence of snow accumulation spatial heterogeneity on snowmelt onset timing and snowmelt rates (Egli et al., 2012; Luce et al., 1998; Lundquist and Dettinger, 2005; Pflug and Lundquist, 2020). Here, we acknowledge that this approach operates on multiple assumptions (discussed above), all of which could vary in accuracy on grid cell level. However, this approach may also provide the opportunity to implicitly represent the heterogeneity of snow in complex terrain and the fraction of the area that could be more supportive for denning habitat (e.g., Fig. 1). We discuss this more in Section 3.2. Readers should refer to Liston (2004) for more information about the subgrid snow methodology described in this section.

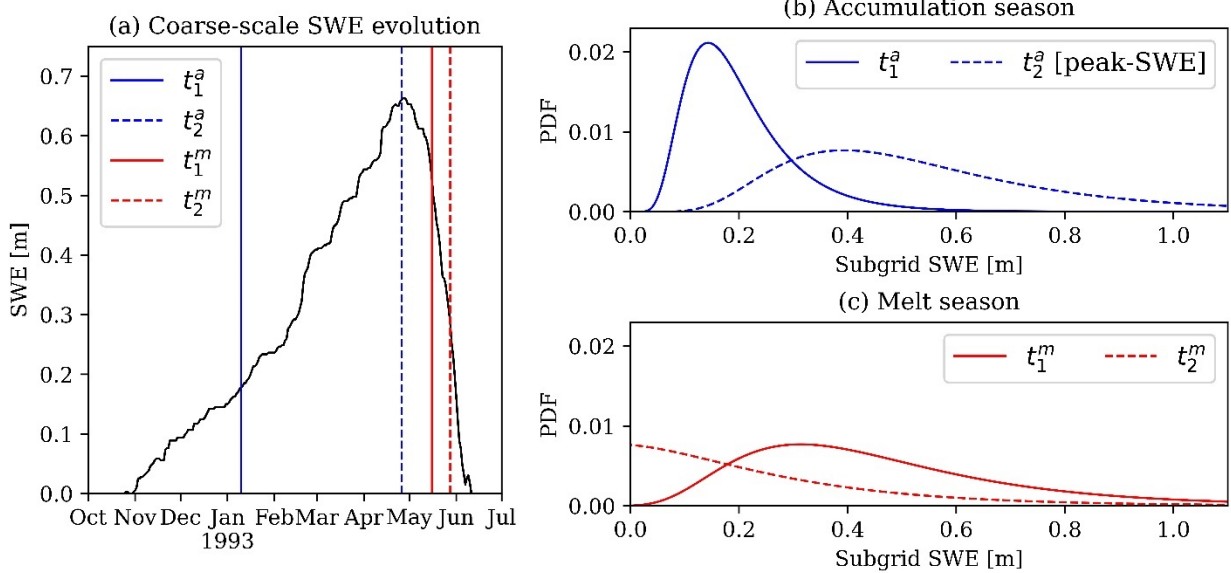

Figure 3. An example of the Liston (2004) subgrid SWE parameterization assuming CoV = 0.5, and SWE evolution for a 480 m grid cell in a random year (panel a). Subgrid SWE distributions are shown for two times ($t$, subscripts 1 and 2) in the accumulation (superscript a) and melt (superscript m) seasons (panels b and c, respectively). The timing of each date corresponds to the matching vertical bar in panel a.

**3.2. Thresholding wolverine habitable area**

The area that could support denning wolverines was calculated for each of the discretizations in each year using a SWE threshold of 0.20 m on 15 May, in accordance with previous studies (e.g., Barsugli et al., 2020; Copeland et al., 2010; McKelvey et al., 2011). For the D480 and D90 discretizations, each cell's denning fraction (DF) was classified as fully-suitable for denning (DF = 1.0) or unsuitable (DF = 0.0) if the 15 May grid cell SWE was greater than or less than 0.20 m, respectively. For the S480 discretization, DF was calculated for each grid cell using:

$$DF = \int_{SWE_m+\beta}^{\infty} f(SWE)dSWE,$$

$$(5)$$

which represented the portion of the cell's SWE distribution greater than the SWE threshold ($\beta = 0.20$ m). PWDA was calculated for each discretization as the sum of DF (in space), multiplied by grid cell area.

Relative to DF calculated from a discrete 480 m grid cell (D480), DF calculated over the same area from the finer-scale discretizations (S480 and D90) could have one of four possible relationships. First, the mean SWE of the D480 grid cell, and the finer-scale distribution of SWE (S480 and D90), could both be entirely greater than the 0.20 SWE threshold. This results in a fully-suitable denning fraction (DF = 1.0) for all discretizations (Fig. 4a). DF would also agree in regions where all discretizations have SWE below 0.20 m (Fig. 4d), resulting in no denning opportunities (DF = 0.0). The scenarios shown in Fig. 4b and Fig. 4c are where DF is sensitive to the discretization. Figure 4b shows a scenario where the coarse-scale mean SWE is sufficiently deep enough to be classified as fully-suitable for denning (SWE > 0.20 m), even though some portion of that grid cell contains SWE that is shallower than the SWE threshold. Therefore, using a finer-scale discretization would result in a net loss in DF, the magnitude of which is shown by the red hatching in Fig. 4b. The opposite could be true for instances where coarse-scale mean SWE falls below the 0.20 m SWE threshold, thereby underestimating denning opportunities relative to finer-scale representations that resolve some deeper snow deposits (Fig. 4c, blue hatching). Here, the three reanalysis discretizations (D480, D90, and S480) were provided identical meteorological forcing, and when coarsened to 480m resolution, had SWE that agreed to within 1%, on average on 15 May. Therefore, the degree to which the scenarios shown in Fig. 4b and 4c occur were the drivers of differences to wolverine denning opportunities.

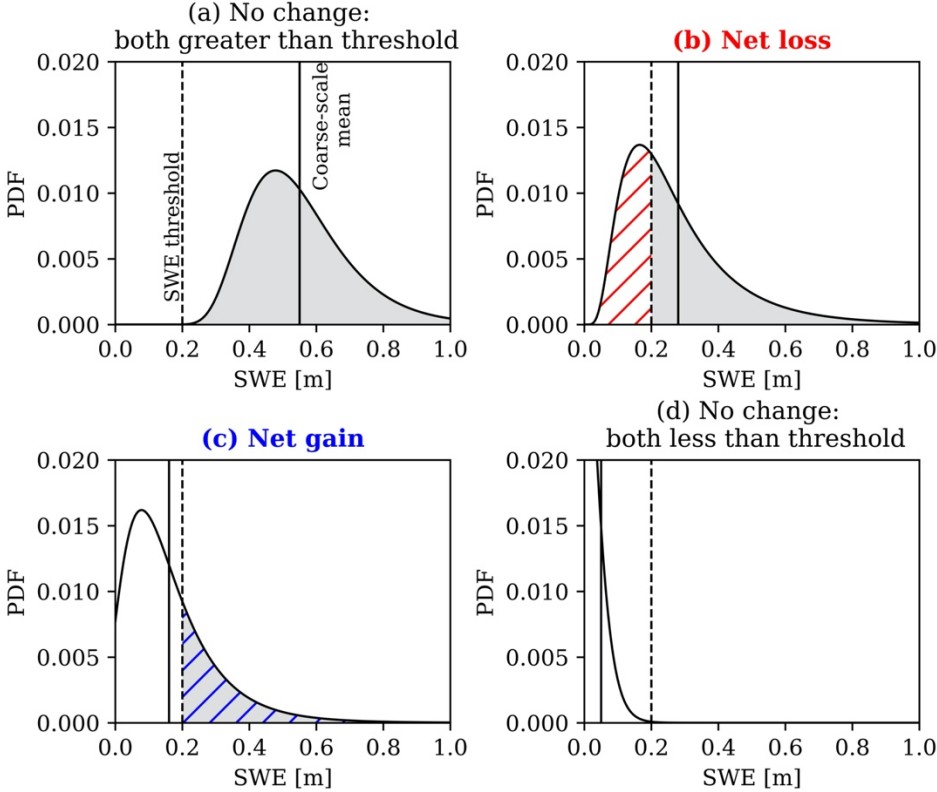


Figure 4. Conceptual portrayal of the similarities (a and d) and differences (b and c) in DF for a 480 m discrete grid
cell (vertical solid line) and a finer-scale representation (distribution) of SWE over the same area. The vertical
dashed lines represent the 0.20 m SWE threshold. Shaded areas show the portion of the distribution with SWE
greater than the threshold. Hatched areas demonstrate differences in DF between the coarser and finer-scale
discretizations of SWE.
**3.3. Categorizing winter climate categories**
To determine PWDA sensitivity to different climatic conditions, we identified years from the reanalysis with
different winter precipitation magnitude and phase (rain versus snow). Here, winter is defined by periods between
October 1st and the date of domain peak SWE volume. Following work from Heldmyer et al. (2023), we used
domain average cumulative winter precipitation and the fraction of the winter precipitation that fell as snow (both
from the reanalysis) as indices for winter precipitation magnitude and the temperature at which precipitation fell.
Using a percentile, we separated years that fell at least that far from the 1985 – 2020 median precipitation magnitude
and fraction of snow precipitation. In doing so, we partitioned years with wet, dry, cold, and warm winter climate
categories. We did this separation using a range of percentiles until the statistical difference (measured using the
Mann-Whitney u-test) in D480 PWDA was maximized between the years with different climatic conditions (warm,
cold, wet, dry, and typical). To avoid spurious results, this percentile was also adjusted to ensure that each climate
category included at least 6 years. This approach maximized the difference in interannual PWDA as a function of
different winter climatic conditions. This was then used as the baseline to compare how much more or less sensitive
PWDA was to the different SWE spatial discretizations.
**4. Results**
Over low-elevation forested grid cells (< 2800 m), SWE accumulation variability was large relative to the smaller
amounts of snow, resulting in large CoV (typically between 0.50 and 0.80) (Fig. 2b and 2c). On mid-elevation
slopes (2800 – 3300 m), CoV tended to be smaller (approximately 0.30, on average). However, CoV increased again
at higher elevations (> 3300 m), and particularly on the leeward side of peaks. This was expected given the more
extreme terrain and increased spatial variability of snow from wind-drifting, preferential deposition, cornice
formation, and avalanching.
The difference in PWDA was maximized between 1) warm and cold years, and 2) wet and dry years, that had winter
precipitation magnitude (Fig. 5a, x-axis) and precipitation phase (Fig. 5a, y-axis) that fell above the 77[th] and below
the 23[rd] percentiles ($\pm$27[th] percentile from the median). These climate conditions had impacts on the evolution of
SWE and snow-covered area (Fig. 5b and Fig. 5c). On average, as compared to years with normal winter
precipitation magnitude and phase (Fig. 5a, white region), cold years and wet years had peak SWE volume that was
23% and 28% greater, respectively. This was opposed to warm years and dry years, with peak SWE volume that was
21% and 31% smaller, on average, than typical water years. The timing of peak-SWE was driven most by the
magnitude of winter precipitation. In fact, average peak-SWE timing was 28 days later for wet years than dry years.
Snow disappearance timing (snow-covered area < 200 $km^2$) was also 21 days later for wet years than dry years.
Statistically, the timing of snow disappearance, crucial for wolverine denning habitat, was explained well by the
peak-SWE volume (r = 0.82) and the date of peak-SWE (r = 0.63), both of which were influenced more by winter
precipitation magnitude than temperature.

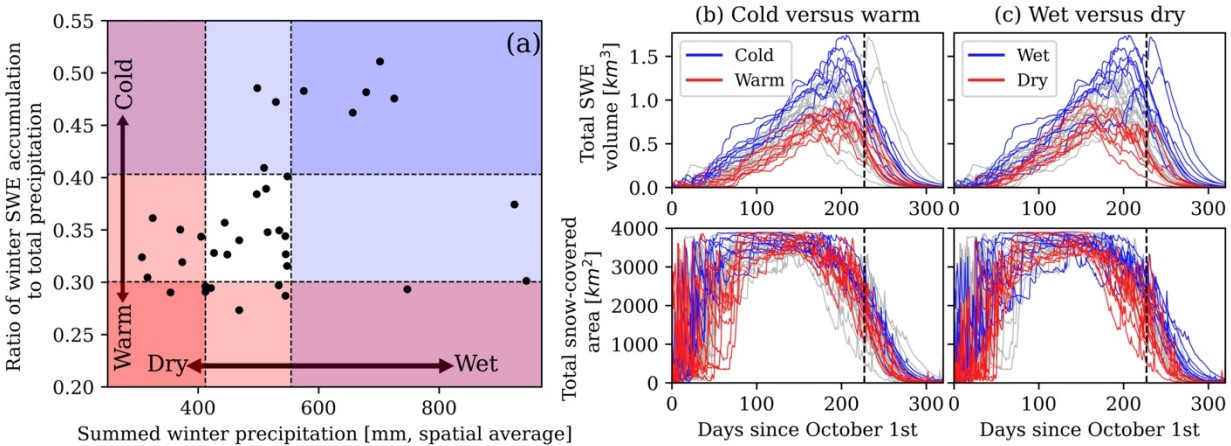

Figure 5. Annual climatic conditions grouped into categories based on winter precipitation magnitude (a, horizontal-
axis) and precipitation phase (a, vertical-axis) outside the 23[rd] and 77[th] percentiles (a, dashed lines). The evolution of
SWE volume and snow cover are compared for warm versus cold (column b) and wet versus dry years (column c).
Vertical dashed lines in columns c and d indicate 15 May.
In all years except dry 2002, PWDA was smaller for the D90 discretization than the D480 discretization (Fig. 6).
This resulted in a 10% reduction to the 36-year median PWDA (Fig. 6b). The PWDA differences between the D480
and S480 discretizations varied more on an annual basis. For years with D480 PWDA less than 1000 $km^2$, S480
discretizations increased PWDA by up to 30%, 11% on average. However, in years with PWDA greater than
1000 $km^2$, S480 PWDA was approximately 3% smaller, on average, than D480 PWDA. In short, the S480
discretization tended to have smaller annual swings in PWDA than the D480 discretization. The causes of these
PWDA disagreements are discussed in Sect. 5.1. Despite the annual differences in D480 and S480 PWDA, the 36-
year median PWDA for these discretizations agreed to within 1% (Fig. 6b).

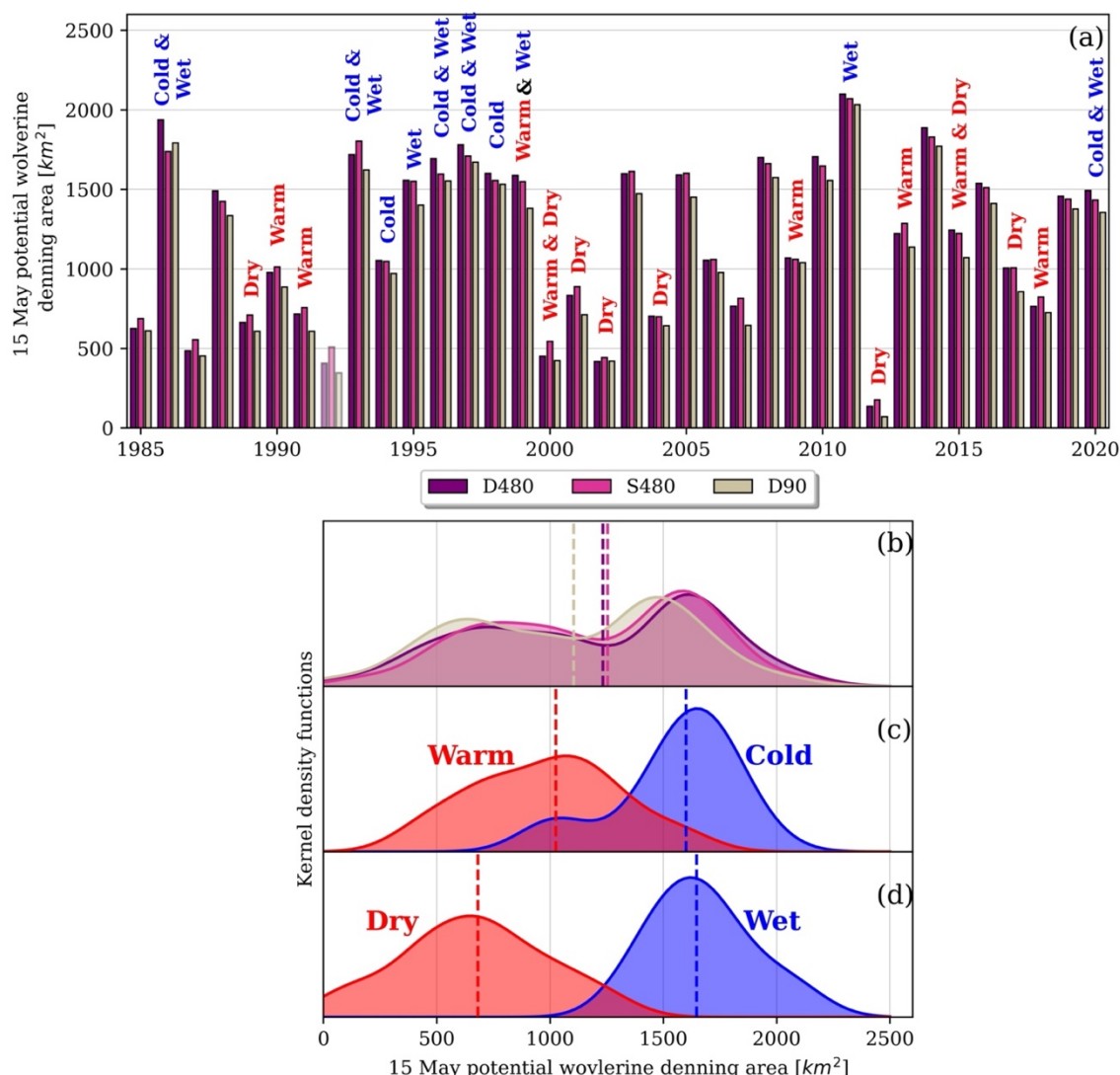


Figure 6. 15 May PWDA compared annually for three different spatial discretizations (a). Lower panels show the
kernel distributions for the data in panel a, separated based on the spatial discretization (b), temperature categories
(c), and precipitation categories (d). The medians of each distribution are shown by the vertical dashed lines (b – d).
The data in panels c and d include data from all three spatial discretizations. The data from WY1992 (a, faded bars)
exhibited artifacts, and was excluded from the kernel distributions (b-d).

Even though PWDA was sensitive to different spatial discretizations (Fig. 6b), PWDA across the 36-year period was
not statistically different between any of the three discretizations (p > 0.48). Conversely, the difference in 15 May
PWDA was significantly larger between the years with different winter climate categories (Fig. 6c and
6d).Differences in PWDA between years with warm and cold conditions were statistically significant (p =
$1 \times 10^{-5}$). Given that 15 May snow covered area were similar between warm and cold years (Fig. 5b), this
difference between warm and cold years in Fig. 6c show that changes to PWDA were driven by changes to SWE
magnitude and the area with SWE exceeding the SWE threshold. Dry and wet years exhibited larger differences to
both 15 May SWE and snow cover (Fig. 5c), resulting in PWDA (Fig. 6d) that was even more different between the
years with these climate conditions (p = $1 \times 10^{-8}$). The impact of these warm, dry, cold, and wet climate conditions
resulted in the bimodal distributions in PWDA shown for the different discretizations across the full time period
(Fig. 6a). While PWDA was not statistically different between cold and wet years (p = 0.34), the distribution of
PWDA in dry years was significantly smaller than the distribution of PWDA in warm years (p = 0.001), showing
that PWDA was more sensitive to conditions that reduced snow habitat, like warm and dry conditions.

The results from Fig. 6 suggested that changes in PWDA across annual periods of differing climatic conditions, or
across future periods with expected changes in climate (e.g., Barsugli et al., 2020) should be informative from a
species status assessment perspective, regardless of the snow spatial discretizations that we tested here. However, as
noted above, the S480 discretization increased PWDA by 11% on average in low snow years, with increases as large
as 30% for individual years. These low snow years often corresponded with drier and/or warmer winter conditions,
the latter of which are expected in the future. For example, the average air temperature during December, January,
and February precipitation events during warm years in the reanalysis record was approximately 0.8° higher than
winter precipitation events in typical years. These conditions are consistent with what is projected for this region by
2055 (Eyring et al., 2016; Scott et al., 2016). This suggests that the disparity between habitat inferred from discrete
grid cells, and grid cells with subgrid snow heterogeneity, could be of greater importance for future snow habitat
assessments. Additionally, using PWDA as the sole metric for evaluating differences in annual opportunities for
wolverine denning may oversimplify the degree to which static thresholds and different spatial discretizations
interact. For instance, PWDA inferred on a static date (15 May) compares very different regimes of the snow season
as wet years had peak SWE timing, and snowmelt season onset, that was 21 days later than typical snow seasons
(Fig. 5). Since shallower snow melts more readily than deeper snow (provided the same energy), comparing SWE
on a static date in years with very different conditions neglects the different rates of habitat depletion for a few days
on either side of the date threshold. These issues are investigated more in Sect. 5.
**5. Discussion**
In this section we diagnose the causes for disagreements in the frequency and locations at which 15 May SWE
exceeded the 0.20m SWE threshold between the three spatial discretizations of snow (Sect. 5.1). We also investigate
how the use of a static SWE threshold and threshold date, may obscure the picture of interannual changes to
wolverine denning habitat availability (Sect. 5.2). Using these findings, we discuss how information provided from
multiple spatial discretizations could provide information about the fidelity and uncertainty of thresholds, as well as
the interactions and tradeoffs between spatial discretizations and thresholds, both in context for assessing snow-
adapted wildlife habitat, and more broadly for other environmental studies (Sect. 5.3).
**5.1. Spatial differences in DF**
The spatial difference in DF between the three discretizations had annually similar patterns, with the largest
differences at locations where the domain had SWE that was near the 0.20 m SWE threshold. This is shown in
Fig.7d and Fig. 7e where the spatial DF disagreements that spiked on 15 May 2008 were focused between
approximately 2800 and 3200 m of elevation. Relative to the D480 discretization, the S480 discretization tended to
increase DF in grid cells at lower elevations where mean SWE was less than the SWE threshold, but some portion of
the grid cell had SWE deep enough to exceed the threshold (e.g., Fig. 4c). The opposite effect occurred at higher
elevations where mean SWE exceeded the SWE threshold, but the lower-tails of the S480 SWE distributions were
below the threshold (e.g., Fig. 4b). As a result, the S480 discretization had a more-gradual increase in thresholded
denning availability with elevation, and a downward shift in the elevations that could support denning wolverines
(Fig. 7f). In fact, relative to the D480 discretization, the S480 discretization had 23% less interannual variability in
the elevation at which equal PWDA existed at higher and lower elevations (Fig. S2a). This was a result of the
subgrid representations of SWE heterogeneity which allowed for gradual and fractional ($0.0 \leq DF \leq 1.0$) increases
in DF with increases in SWE. This was opposed to the D480 discretization, which could only resolve binary DF (0
or 1 for SWE less than and greater than 0.20 m), resulting in larger elevational shifts in the annual locations that
could support wolverine denning.

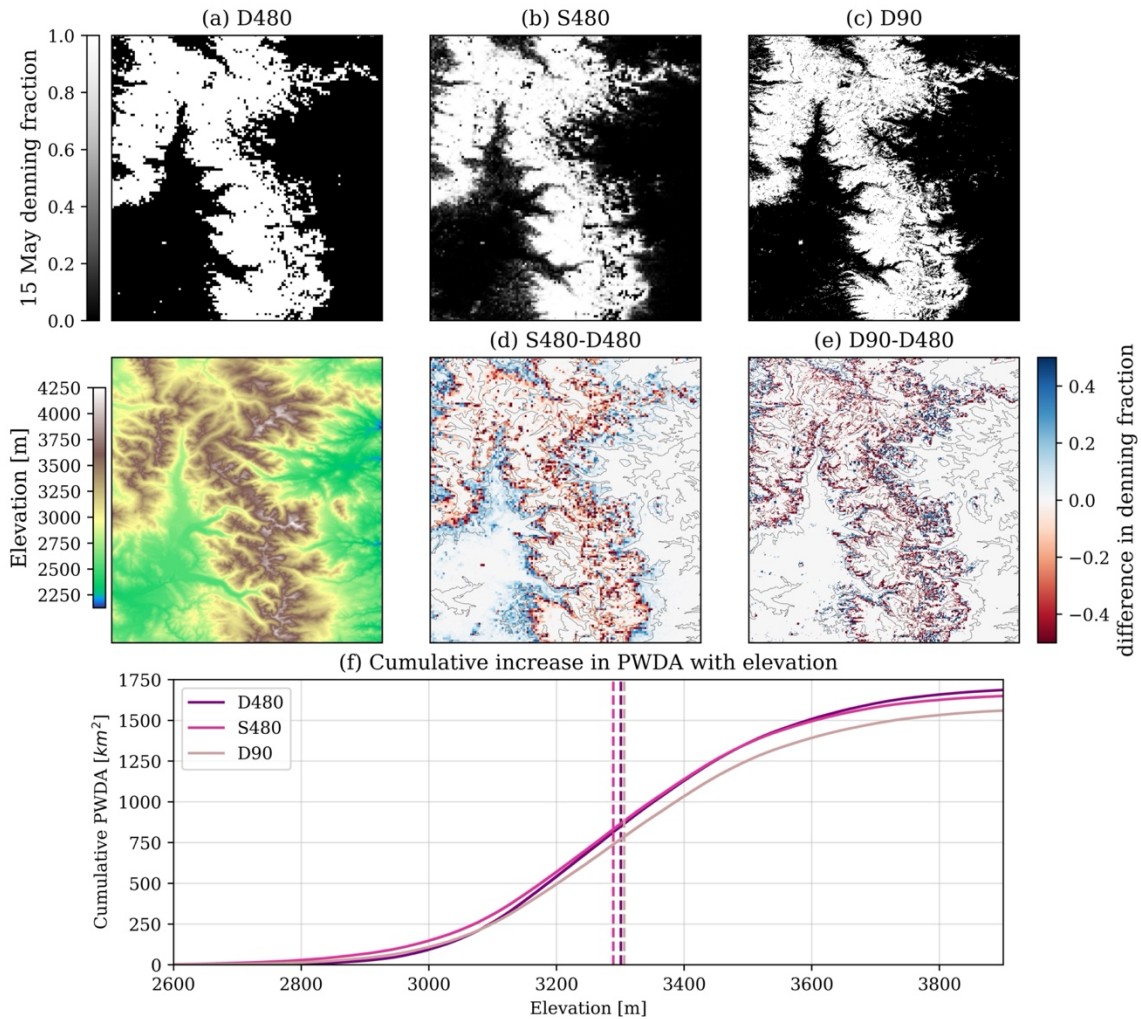


Figure 7. Spatial comparisons of DF for the three discretizations on 15 May 2008. Panel f compares the cumulative
PWDA (y-axis) calculated for grid cells sorted in order of increasing elevation (x-axis). Vertical dashed lines show
the elevation of median PWDA, or elevation at which PWDA is equal for higher and lower elevations.
Relative to the D480 discretization, the D90 discretization also tended to increase DF at lower elevations. However,
all years had reduced D90 DF in elevations higher than approximately 3120m. This was the cause of the 10%
reduction in D90 PWDA, relative to the other discretizations (Fig. 6b). These decreases were typically located on
unvegetated, exposed, and steep slopes, where it was likely that winter snow retention was decreased, snow
sublimation was increased, and sloughing to lower-elevations was more common (Bernhardt and Schulz, 2010;
Grünewald et al., 2014; Machguth et al., 2006). This demonstrates the utility of the observation-based reanalysis
used in this study, which may have resolved thinner snow deposits on slopes with decreased snow retention and/or
enhanced snow removal by processes like sloughing, both of which are among the most-difficult processes to
represent with models. The D480 discretization averaged snow from surrounding areas, smoothing out thinner snow
deposits resolved by the D90 discretization. Although attempting to resolve subgrid snow heterogeneity, the
evolution of SWE assumed by the S480 simulation, which assumed lognormal snow accumulation and spatially-
uniform subgrid snowmelt (Fig. 3), may have been less-appropriate for the areas containing these isolated thinner-
snow 90 m grid cells. While the D90 discretization decreased total PWDA, D90 snow cover was also patchier (Fig.
7c), which could also influence the movement and connectivity for wolverines (USFWS, 2018) and other snow-
adapted species.
Winter precipitation magnitude and temperature influenced the volume of snow and the elevation of the snow line
that existed on 15 May in each year. Since the differences in DF between the discretizations were largest at grid cells
near the 0.20 m SWE threshold, often located just above the snow line, the spatial pattern of DF differences (e.g.,
Fig. 7) exhibited an interannually-repeatable relationship with the dry, warm, cold, and wet winter climate categories
(Fig. 5). To show this, we calculated the differences in DF between all three discretizations (D480 versus S480,
D480 versus D90, and S480 versus D90) in all 36 years. Then, for each 480 m grid cell, we identified the climate
category that resulted in the greatest mean absolute differences in DF across the three discretizations. The climate
categories that had the greatest influence on DF uncertainties covered similar portions of the domain, with 33.7%,
20.9%, 25.2%, and 20.2% being most attributed to dry, warm, cold, and wet conditions, respectively (Fig. 8). At low
elevations (2650 – 3050 m), 15 May snow typically existed only in wet years. In those years and elevations, mean
SWE for the D480 and D90 discretizations often fell below the 0.20 m SWE threshold. However, the large CoVs of
subgrid SWE accumulation in these elevations (Fig. 2) resulted in S480 subgrid SWE distributions with upper-tails
that sometimes exceeded 0.20 m (e.g., Fig. 4c) (Fig. 8c). This was in-line with findings from Magoun et al. (2017),
who noted suitable denning conditions at lower-elevations, even in instances when the surrounding terrain was
predominantly snow-free.
The average differences in DF between the three discretizations were largest in cold years for elevations spanning
3050 – 3150 m, and in warm years for elevations spanning 3150 – 3350 m (Fig. 8). Across this elevation range
(3050 – 3350 m), both of the 480 m discretizations (D480 and S480) estimated more denning opportunities than the
D90 discretization (Fig. 8c). However, at higher elevations (> 3350 m), DF calculated from the S480 discretization
approached DF calculated from the D90 thinner snow deposits (Fig. 8c).

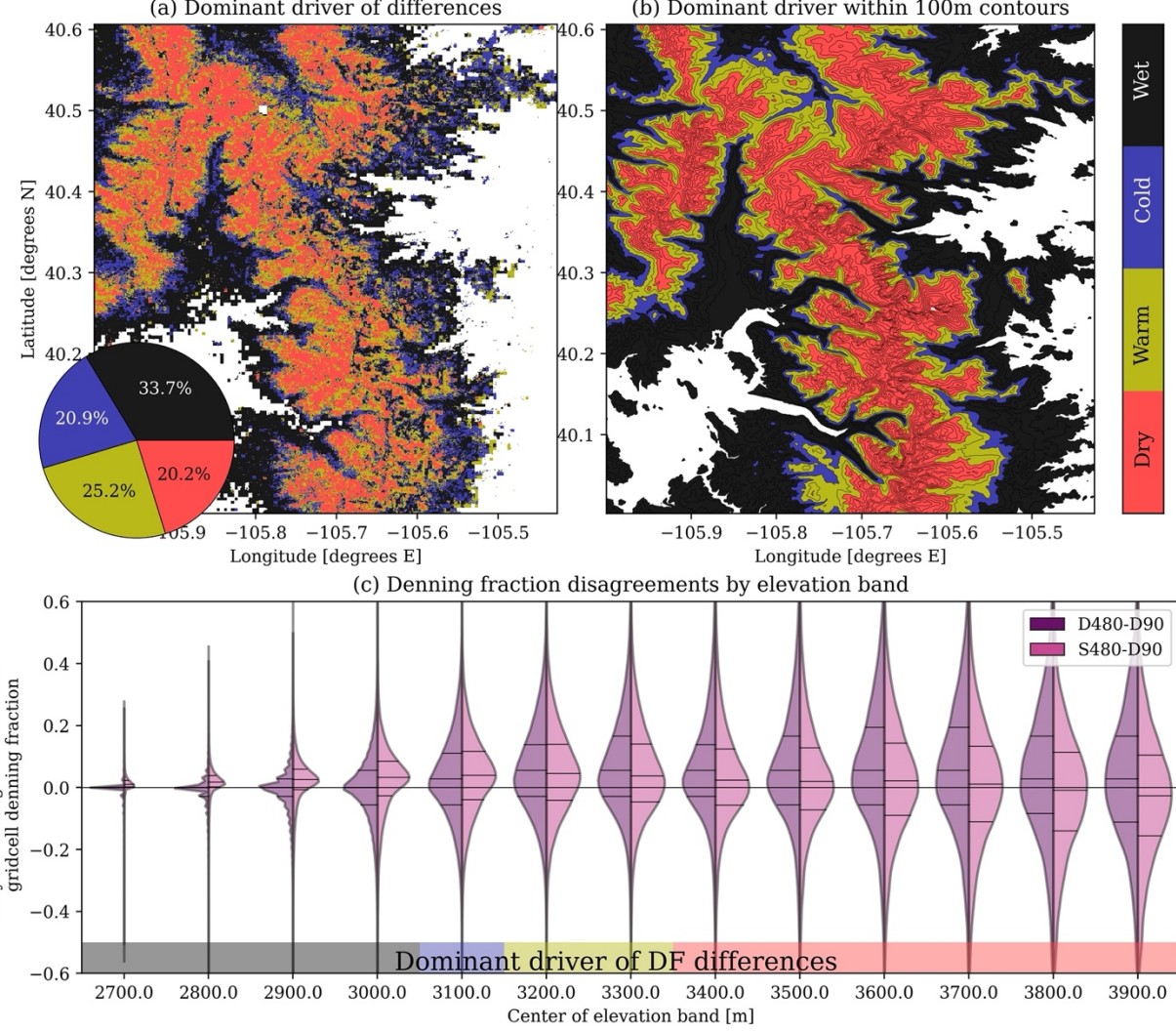


Figure 8. Winter climate categories that most-influenced DF disagreements between the three discretizations (a).
Panel b shows the most-prevalent influence from panel a, for 100 m elevation bands. Using DF from the D90
discretization as a reference, the 36-year average difference in DF for the D480 and S480 simulations are shown by
distributions for each 100 m elevation band (c). Lines inside the distributions show the median and interquartile
range.

## 5.2. Threshold sensitivities

To this point, we assumed confidence in the SWE (0.20 m) and date (15 May) thresholds. However, small changes
to either threshold could influence annual estimates of PWDA (e.g., Copeland et al., 2010; Magoun et al., 2017). In
Fig. 9, we show PWDA calculated from a range of realistic SWE thresholds and threshold dates. The range of SWE
thresholds (0.20 ± 0.07 m) were determined using a snow depth of 0.50 m, corresponding to observed wolverine
dens (USFWS, 2018), and the 90[th] percentile range of 15 May snow densities from SNOTEL observations (Fig. 2a)
between 1985 and 2020 ($260 - 540\ kg/m^3$). The range of threshold dates spanned a period of ± 2 weeks,
corresponding to the difference in peak-SWE timing between dry and wet years (Fig. 5). This month-long time span
is also consistent with the observed range of wolverine birth dates (Inman et al., 2012). PWDA sensitivity was
calculated using all combinations of SWE and date thresholds, both of which were discretized at 14 equally-spaced
increments (Fig. 9, left). Then, the gradients (direction and magnitude of greatest change in PWDA) were calculated
from each unique combination of SWE and date thresholds. The gradients were summed using vector addition (Fig.
9, right column) to determine 1) the total rate of change in PWDA with changing thresholds (arrow length), and 2)
the degree to which PWDA was sensitive to one threshold versus the other (arrow angle). This process was repeated
for each discretization and year.

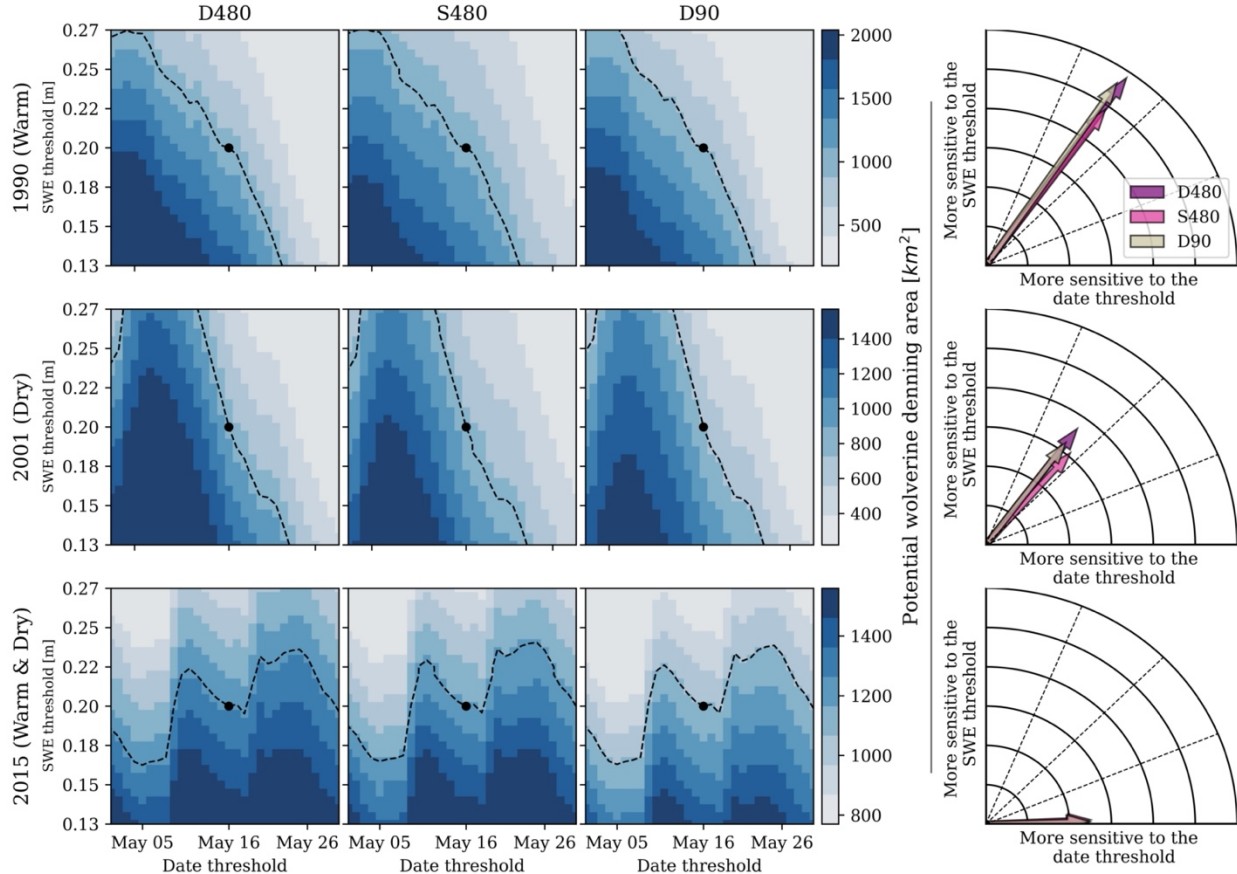


Figure 9. PWDA calculated using different SWE (y-axes) and date thresholds (x-axes), for the different
discretizations (columns), in three different years (rows) with very different sensitivities. PWDA calculated from the
default thresholds (0.20 m SWE on 15 May) is shown by the black circle. Combinations of thresholds that could
reproduce the default PWDA are approximated by the dashed contour. The rightmost arrows show the total direction
and magnitude of PWDA changes with changes in the thresholds.
PWDA in warm 1990 was 18% more-sensitive to the SWE thresholds than the threshold dates (Fig. 9, top row). To
put this another way, the change in PWDA across a period of ± 3 days from 15 May was approximately equal to the
change in PWDA from adjusting the SWE threshold by ± 2.5 centimeters. This sensitivity was similar to the
average threshold sensitivity from the 36-year reanalysis record (Fig. S2b). However, multiple years exhibited
unique sensitivities. For example, spring snowfall between 1 May and 6 May 2001 (Fig. 9, middle row) caused
PWDA to both increase and decrease over the range of date thresholds (assuming a constant SWE threshold).
Therefore, PWDA changed based on whether the threshold date was before, during, or after the May snowfall event,
buffering the degree to which thresholded denning habitat estimates were influenced by the specific winter
meteorological conditions that occurred in that year. This effect also occurred in 2015, when 15 May fell between
two spring snowfall events (Fig. 9, bottom row). As a result, PWDA tended to increase, on average, over the range
of threshold dates, resulting in heightened sensitivities to the date on which denning opportunities were evaluated.
These spring snowfall events had large impacts on 15 May PWDA, but are unlikely to accurately represent the
habitat opportunities and stresses that wolverine were subject to in that year. This demonstrates the dangers of
thresholds applied on static dates, and suggest that metrics over multiple dates (e.g., number of May days exceeding
a SWE threshold) and across sequences of years could be more accurate representations of snow refugia.
PWDA varied by as much 82% between the realistic thresholds shown in Fig. 9. This was similar in magnitude to
the differences in PWDA between years with opposing winter climate anomalies (Fig. 6c and 6d). Across the years
evaluated in this study, the sensitivities to the thresholds were largest for the D480 simulation, and smallest for the
S480 simulation (Fig. 9 and Fig. S2b). As discussed in Sect. 5.1, the S480 discretization, which represented subgrid
snow distribution and fractional changes to DF with changes to the SWE threshold and threshold date, had less
sensitivity to annual changes in meteorological conditions. Similarly, small changes in the SWE threshold and
threshold date changed the prevalence of snow that exceeded the static threshold for discrete grid cells by larger
amounts than the S480 discretization. This suggests that studies with subgrid representations of snow heterogeneity
may decrease the overall sensitivity to SWE and date thresholds.

## 5.3. Threshold caveats and future suggestions

The D90 and S480 discretizations provided unique, but different advantages for estimating PWDA. We believe that
the upper-elevation decreases in D90 SWE and denning habitat on steep and unvegetated surfaces were realistic.
These results were contrary to the findings from Barsugli et al. (2020), who in the same domain, found that finer-
scale physically-based simulations resulted in net increases in wolverine denning opportunities. However, this
analysis used a joint model and observation-based approach (Sect. 2) that may have implicitly represented decreased
snow retention and/or snow sloughing better than the physically based models used by Barsugli et al. (2020). The
discretization with subgrid snow heterogeneity (S480), which is not as commonly used, had less-dramatic swings in
PWDA with changes in annual winter climatic conditions (Fig. 6) and thresholds (Fig. 9). We therefore think that
subgrid representations of snow may be important for habitat assessments, especially given that snow deposits
suitable for denning at scales of 10 m or less sometimes occur in regions with otherwise little snow (Magoun et al.,
2017).

The results of this study suggest that uncertainties provided from combinations of multiple discretizations, applied
across a range of realistic thresholds, would be more informative than a single discretization and set of thresholds.
For instance, SWE volume on 15 May 2015 was 10% less than the 36-year median 15 May SWE volume. However,
due to spring snowfall (Fig. 9), SWE volume on 30 May 2015 was 31% greater than the 36-year median on the same
date. Multiple discretizations could also be used to identify the locations of most (e.g., Fig. 4a and 4d) and least-
certain (Fig. 4b and 4c) opportunities for denning habitat. This information could be used as the basis for identifying
the locations where remote sensing or field campaigns could hone annual estimates of refugium, given that year's
meteorological conditions. Altogether, differences across discretizations (e.g., Fig. 6) and threshold sensitivities
(e.g., Fig. 9) could also be used to provide uncertainty bounds for PWDA calculated in any given year.
Our results show that caution is warranted when combining gridded data and static thresholds. While we focus on
the impact that thresholds and different snow spatial discretizations have on approximations of wolverine denning
opportunities, we expect these results to be applicable to other environmental applications. For instance, while
temperature thresholds are widely used to partition rain and snow precipitation in models, temperature discretized at
different spatial scales could influence the spatial variability of temperature and resulting snowfall volume
thresholded across one or many snowfall events (e.g., Jennings et al., 2018; Nolin and Daly, 2006; Wayand et al.,
2017). Snow cover thresholded using visible and infrared satellite observations may also require changes based on
the size of the satellite pixels and the underlying topographic and vegetative characteristics (Härer et al., 2018;
Pestana et al., 2019). Future studies should report the extent to which different spatial discretizations and ranges of
realistic thresholds influence results. This information could be used to report the 1) uncertainty of thresholded
outputs, 2) fidelity of different gridded products, and 3) the degree to which multiple spatial discretizations could be
combined to improve the fidelity and transferability of results.
**6. Conclusions**
Potential wolverine denning area (PWDA) was thresholded using a published SWE threshold (0.20 m) on a
threshold date (15 May) in a Colorado Rocky Mountain domain between 1985 and 2020. Results showed that
PWDA was statistically different ($p < 0.01$) between years with different winter precipitation magnitude (wet versus
dry) and precipitation temperature (cold versus warm) conditions. In fact, climate-driven differences in annual
PWDA were substantially larger than differences in PWDA between snow discretized using 1) discrete 480 m grid
cells, 2) 480 m grid cells with subgrid representations of SWE heterogeneity, and 3) discrete 90 m grid cells.
Therefore, studies that assess changes in habitat health for species like wolverines with past and future changes in
climate could be informative, regardless of the spatial discretizations tested.
Despite the sensitivity to winter climatic conditions, annual differences in denning patterns and parameter
sensitivities emerged for the different discretizations. For instance, 90 m grid cells resolved thinner snow deposits in
mid-to-upper elevations (approximately 3050 – 3350 m) that were not resolved by either of the 480 m
discretizations, decreasing PWDA by 10%, on average. Snow discretized with subgrid representations of SWE
spatial heterogeneity also had less-dramatic swings in annual PWDA. The simulations with subgrid SWE
heterogeneity increased PWDA by 10 – 30% in low-snow years, many of which were representative of future
changes in average temperature expected over the next 50 years. Spatially, the differences in the prevalence of SWE
that exceeded the threshold between the three different snow discretizations were heightened at the grid cells that
had SWE values close to the SWE threshold (0.20 m) on 15 May, the elevation of which was driven in large part by
the winter climatic conditions. On average, PWDA was more sensitive to the SWE threshold than the date threshold,
but had the smallest amount of sensitivity to the 480 m simulation with subgrid snow heterogeneity, which had more
gradual changes to the fraction of a region exceeding the SWE threshold with small changes in SWE. This
discretization also had the least amount of sensitivity to interannual changes in winter climatic conditions. However,
some years had late-spring snowfall events, altering the amount of PWDA by up to 82% depending on whether the
threshold date was before, during, or after the snowfall event.
Our results show that differences in how snow is spatially discretized can influence information generalized using
thresholds. Therefore, future studies thresholding spatiotemporal environmental data should include multiple spatial
discretizations and ranges of realistic thresholds to provide a more comprehensive picture of uncertainties associated
with chosen thresholds and datasets. Although we used wolverine habitat as an example, we expect these results to
be applicable to any study thresholding environmental data, especially for studies generalizing information at spatial
scales finer than those of modeled or observed resolutions.
**Code and data availability**
Readers are encouraged to enquire about the most up-to-date version of the reanalysis from the principal developer,
Steven Margulis. Scripts used in this manuscript are provided at https://github.com/jupflug/HABITAT-
threshold_vs_discretization.
**Author contributions**
JP and BL designed the experiments. YF and SM provided the snow reanalysis. JP wrote the manuscript, with
comments provided from all authors, and special supervision by BL.
**Competing interests**
The authors declare that they have no conflict of interest.
**Support**
This work was supported by the CIRES Visiting Fellows Program, the United States Geologic Survey (USGS)
award G21AC10645, and the National Integrated Drought Information System (NIDIS) Grant NA20OAR4310420.
**Acknowledgements**
We would like to thank and acknowledge support from current and past U.S. Fish and Wildlife Service staff, in
particular, John Guinotte and Steve Torbit.

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
