# Peer review of "Short summary. Wolverine denning habitat inferred using a snow threshold differed for three different spatial"

_EGUsphere, 2022_

## Author Response (AR1)

Response to RC1.

We would like to thank Anonymous Reviewer #1 for their thoughtful and constructive feedback. We were particularly appreciative of their comments on wolverines and their denning habitat, which helped us improve the terms and text that we used to discuss the connection between snow resources and denning refugium.

With respect to the major comments, we agree that our use of "wolverine habitable area (WHA)" could imply that regions exceeding the SWE threshold, and only those regions, would support denning. As Reviewer # 1 notes, the disconnect between snow at denning spatial scales (< 10 m) and gridded representations of snow at coarser resolutions (90 – 480m) serves as a motivator for this study. Therefore, we take the reviewer's recommendation, replacing WHA with "potential wolverine denning area (PWDA)". We also agree that our results and discussion frequently flipped back and forth between descriptions and acronyms (D480, S480, D90, HF, and WHA), making the text difficult to follow at times. We made every effort to ensure that acronyms are now defined early and used consistently throughout.

Both reviewers expressed a need for more information on the SWE reanalysis. Therefore, we expanded on Section 2.2 of this manuscript We also added a passage noting that the snow cover observations used for this reanalysis were from Landsat remote sensing. Prior research by coauthor Margulis also developed an approach for integrating snow cover observations from both MODIS and Landsat remote sensing (Margulis et al., 2019), but found little to no improvement in SWE estimates due to the coarse-resolution of MODIS observations in mountainous terrain. We also edited the text in Section 2.2 to make it clear that SNOTEL data was not assimilated or used in the reanalysis, but is instead an independent data source that we used to evaluate the accuracy of the reanalysis.

Finally, although the principal reanalysis developers are coauthors on this study (Margulis and Fang), the SWE reanalysis that we used is a popular product that is seeing widespread use in the snow community (e.g., Fang et al., 2020; Ma et al., 2023; Pflug et al., 2022; Yang et al., 2021). Additionally, the methods we use are not specific to this product, but are instead intended to highlight the interactions between thresholds and gridded products that could be present for any snow dataset. In our edits we tried to strike the balance between providing the information that is necessary to understand this product, and providing the sources for more nuanced information if readers want to learn more.

Detailed comments are responded to below. Comments from the reviewer were grouped if they could be responded to together. Line numbers below reference the manuscript with tracked changes.

**Detailed comments:**

*Line 26 – Does "Relative to the 480 m grid cells" refer to both (1) and (3) in the previous sentence, or only (1)?*
*Lines 28-30 – Does "grid cells with subgrid representations of snow heterogeneity" only refer to (3) on lines 25-26?*

*Line 31 – Does "simulations without subgrid snow heterogeneity" refer to (1) and (2) on line 25?*

> Thanks for pointing this out. The first case refers to both the D480 and S480 discretizations (edited in lines 30-35). You are correct in your understanding of the next two sentences. Your suggestion about more consistent use of the acronyms improved the clarity of this passage and the full manuscript.

*Lines 61-62 – Why limit this to North American wolverines? This misleadingly implies a different association with snow among palearctic wolverines, for which there is no evidence. Please change to "wolverines (Gulo gulo)".*

> Good point. This text was edited to be more inclusive (lines 63 - 68).

*Line 64 – Several of the studies cited in the previous sentence, and several not cited, have explicitly identified this problem and attempted to address it by representing snow at resolutions expected to be biologically relevant. Consider citing a few such examples here: "(but see Mahoney et al. 2018, Glass et al. 2021, Liston et al. 2016)."*

> This is a good idea. These citations were added in lines 68 - 69. Thanks for passing along the Glass et al. 2021 citation.

*Lines 77-80 – Please provide more detail regarding where these thresholds (SWE and date) originated. At what spatial and temporal resolution were they originally linked to wolverine denning? How many dens were used to make this association? Which of the cited studies identified these thresholds? Given this paper's focus on the importance of transparency regarding thresholds, it seems particularly important to address spatial and temporal resolution these thresholds are thought to represent, and therefore any reason they may result in biased estimates of wolverine denning area when applied to other spatial discretizations.*

> We agree that the original text didn't do a good enough job at identifying where the SWE threshold and threshold date originated from. We edited the text to improve the clarity, and split the portions of the text and corresponding references for the SWE threshold and threshold date (lines 80 – 91).

*Line 83 – Consider replacing "wolverine habitat" with "wolverine dens" to avoid conflating the two. "Wolverine" does not need to be capitalized here.*
*Lines 156 – Consider replacing "wolverine habitat" with "denning wolverines" (also at lines 222, 347, 393).*
*Lines 352-353 – Please replace "annual location of wolverine habitat" with "potential wolverine denning area" or similar (see comment on line 232).*
*Line 495 – Please replace "wolverine habitat" with WHA or PWDA.*
*Line 499 – Please replace "wolverine habitat" with WHA or PWDA.*

> Good catch. We checked and revised our use of the terms "habitat" and "dens" throughout the manuscript.

*Lines 92-93 – Consider "We focus on"*

We agree that this reads better (lines 100 – 101). Thanks for the suggestion.

*Lines 114-115 – Please clarify that this region does not currently support a reproductive population of wolverines.*

We added this point to the sentence in lines 126 – 128.

*Lines 117 – Comma in "Fig, 2."*

This text was edited (line 130).

*Line 123 – It is unclear why this section (SWE Reanalysis) is included under "Domain and data" rather than "Methods"; this presents meaningful data processing and I would generally expect it to appear under methods, though perhaps this is a journal-specific guideline. I would defer to the editor on this.*

Thanks for your comment. We did not make it clear enough in the original text that this SWE product was not a major focus of our methods, but instead a product that has seen widespread use in the snow community that we accessed for this study (see our response to the major comments above).

We also agree that the latter half of this section, which touches on how we estimated SWE subgrid variability is more of a method. We attempted to move this in the most recent manuscript. However, we found that it's previous layout which introduces the CoV, but points to the methods in Section 3.1, was the easiest to comprehend for colleagues that were able to review the revised manuscript.

*Lines 124-139 – What satellite data are used here? "Satellite-observed snow cover disappearance" is mentioned, but it is unclear what the source of these data are, or what spatiotemporal resolution they represent.*

The product uses Landsat observations of snow cover evolution. This was added to the text. See our comment on this in the responses to the major comments above.

*Lines 193-194 – Please reiterate here that CoV is derived from reanalysis-based SWE values at 90 m resolution for overlapping pixels.*

This is a great suggestion. It was added to the text (lines 220 – 222).

*Lines 244 – At what spatial and temporal resolution do the three spatial discretizations "resolve similar SWE"?*

We edited this text to clarify that we did comparisons between these discretizations by aggregating to the common 480m grid (lines 278 – 282).

*Lines 267-268 – I would recommend removing or clarifying this sentence; "had a relationship with terrain" is too vague to convey meaning.*

    We agree. This sentence was removed.

*Lines 272-273 – This would seem to imply that the 90 m reanalysis product is capable of simulating these processes (wind-drifting, preferential deposition, cornice formation, and avalanching), but I don't expect that's actually the case for the model developed by Xue et al., 1991. Please remove or clarify, and expand the description of the Xue et al. (1991) model used here.*

*Lines 364-367 – More information is needed regarding the combination of models and observations used for reanalysis (as indicated by my confusion on lines 272-273 as well). Please expand section 2.2 (SWE Reanalyses) to more thoroughly present the model used to simulate SWE based on meteorological reanalysis forcing. It is unclear based on lines 124-149 how the modeling process used here could account for processes like sloughing and avalanches.*

    You are correct that the Xue et al. (1991) model is incapable of explicitly representing these processes. However, the snow reanalysis uses an ensemble of simulations with different amounts of accumulation, subgrid SWE variabilities, and melt characteristics. It then reweights those simulations to favor the ensemble members that capture fractional snow cover evolution as observed by Landsat. In our past research (Pflug et al., 2022), we have seen that this approach, which adjusts the peak SWE accumulation, rate of snow depletion, and snow heterogeneity implicitly captures the combined impacts from complicated processes like those listed above. We added a brief discussion of this in Section 2.2 (see response to major comments above).

*Lines 321-323 – This is a valuable point, and I think it can be taken one step further. Since the 11% mean WHA increase by S480 in low snow years is never matched by the D90 WHA estimates, it would seem not only that this difference will be "of greater importance," but that the subgrid methodology will increasingly overestimate WHA as the climate continues warming. That is, S480 is biased high for low-snow years (if we assume D90 as the most accurate estimator of WHA considered here), and that bias will only become more pronounced with continued warming.*

    Thanks for your comment. While we believe that the D90 simulation may represent the "patchiness" of snow at high elevations better than the D480 simulation, we don't have the evidence to determine which of the three discretizations have the most-valid representation of potential wolverine denning area (PWDA). Therefore, we would hesitate to say that PWDA is biased high for the S480 discretization. Based on our experience in this region, it is not uncommon to see isolated, and small-scale (< 10 m) snow deposits that persist late into the snowmelt season when all surrounding areas are snow-free. Magoun et al. (2017) also suggested that these sorts of snow deposits can support denning. This manuscript doesn't attempt to identify the best discretization, but stresses that approaches that use thresholds should be aware of these uncertainties that can result from different discretizations, and do their best to report and investigate them.

*Line 323 – Please replace "annual wolverine habitat" with "potential wolverine denning area" or similar. Suitable habitat fluctuates on generational, not interannual, timescales; instead, wolverine habitat in Colorado would likely be defined by having enough years (i.e., some proportion) with sufficient snow for wolverines to successfully reproduce.*

We have replaced WHA with PWDA throughout (see responses to the major comments above). This is also a really good point about the long-term duration of snow that is necessary to support wolverine populations. We also include brief acknowledgements of time-averaged metrics in both Section 5.2 and 5.3.

*Line 325 – Period in "…season. as wet…"*

Thanks for catching this. This typo was corrected.

*Line 327 – I think the authors mean "comparing SWE on a static date," since a static date is inherent to WHA.*

That is correct. This was edited (lines 374 – 377).

*Line 340 – Present tense: "This is…"*

This was corrected.

*Line 349 – Change to "(i.e., the elevation at which equal WHA existed at higher and lower elevations)" if that is what's intended.*

We agree. This text was edited (lines 397 – 400).

*Line 352 – Perhaps "elevational" or "altitudinal" is more accurate than "topographical."*

That's correct. This was edited (lines 401 – 404).

*Line 355 (Figure 7) – It is unclear why D480 is represented twice. I would recommend removing one panel and replacing it with a map indicating elevation for the same area.*

We put D480 in the second row to clarify that we were differencing the D90 and S480 thresholded maps versus the D480 thresholded map. However, we like your idea about including the elevation map instead. This figure was edited.

*Line 360 – Does "higher than the snow line" mean across the entire study area? Please clarify.*

We think that the previous sentence was too vague. This was changed to "elevations higher than approximately 3120m" (line 412).

*Line 372 – "Wolverines" does not need to be capitalized here.*

This was corrected.

*Lines 386-389 – More caution is warranted in presenting the results of S480, particularly at low elevations. Liston (2004) specified that the CoV used to parameterize the lognormal curve should be derived prior to any melting in the grid cell, but the authors imply that mid-winter melting did take place at these elevations, and was likely responsible for driving up the CoV (lines 268-271). I can't help but wonder if applying Liston's method to low elevation grid cells is inappropriate for this reason, and has led to inflated CoV values that consequently overestimate WHA at low elevations. This notion seems to be independently supported by the poor match of S480 and D90 at low elevations. Regardless, since the within-cell variability of S480 is derived from (and therefore essentially a simplification of) D90, it seems inappropriate to present the deviation of S480 from D90 as anything other than an artifact introduced by Liston's lognormal subgrid representation approach, or the inappropriate use thereof.*

This is a great comment. We want to start by noting that the CoV was calculated using the SWE accumulation, or the sum of the positive increases in SWE. Therefore, the CoV is not increased by winter snowmelt. We agree that the text in this passage seemed to imply that mid-elevation CoV was reduced because midwinter melt did not occur. This was removed from the revised manuscript. In warm and/or dry years, it was also uncommon for S480 grid cells to have 15 May SWE distributions that exceeded the SWE threshold at elevations less than approximately 3000m. Therefore, PWDA was not influenced by the higher CoV values at lower elevations in those years (Figure 2b and 2c).

We disagree that the difference between PWDA from the S480 and D90 discretizations are only due to artifacts. There are a few reasons for this. First, although CoV was estimated for each 480m grid cell based on the overlapping 90 m grid cells, a continuous lognormal distribution (Equation 2) attempts to represent the less-common snow conditions within that larger region, like thinner than normal snow deposits (e.g., tree wells, wind-scoured locations, etc.) and deeper than normal snow deposits (e.g., wind drifts, terrain shaded snow, etc.). A continuous lognormal distribution with CoV prescribed from the overlapping 90m grid cells would include SWE edge-cases, like the most-extreme deep and shallow snow deposits, that aren't represented from the 90m grid cells alone. Given these SWE variabilities, it is likely that 90 – 480 m grid cells with mean SWE near the SWE threshold would not contain an area with all portions greater or less than the threshold, but instead some fraction of the grid cell with SWE that exceeds the threshold (e.g., Figure 4b and 4c).

Personal experience in this region and results from Magoun et al. (2017) make us believe that an approach like S480, which allows some fractional subgrid estimation of deeper and shallower snow deposits could be a feasible approach. However, you are correct that the subgrid approach operates on a number of assumptions such as pre-defined theoretical distributions and subgrid spatially-uniform snowmelt rates, which may vary in accuracy across the study domain. We added an acknowledgement of this in lines 243 – 248.

*Lines 409-411 – Consider adding "and is consistent with the observed range in wolverine parturition dates (Inman et al. 2012)"*

> This is a really great suggestion. It was added to the text, but we simplified "parturition dates" with "birth dates" for readers that are less familiar with this term (lines 464 – 465).

*Line 410 – It's unclear what "dates of observed wolverine habitat" refers to. Other studies have proposed threshold dates based on alignment with certain aspects of wolverine biology; please clarify what's intended here.*

> We think that this sentence was more confusing than helpful, and was not necessary for the test discussed in this paragraph. It was removed from the updated manuscript.

*Lines 428-436 – This is a valuable point, and its discussion should be expanded a bit. Specifically, the years experiencing spring snowfall highlight the inappropriateness (in at least those cases) of using a single date threshold to measure habitable area for wolverines. If all snow melts by April 15, but then a spring storm dumps 1.5 meters on May 14, what good is it to wolverines? I suggest that the authors mention a few alternate metrics of snow as denning habitat that could be more useful (i.e., representative of what a wolverine uses/needs) than these thresholds (e.g., number of days between late February (parturition) and early May (den abandonment) when SWE exceeds a certain value).*

> This is a great point. We tried to allude to this, but should have stated this more explicitly in the text. We added this point in lines 492 – 496.

*Lines 442-443 – Please rephrase as "…may decrease the sensitivity to changes in SWE and date thresholds." Including "uncertainties" implies that subgrid representations can be used to account for uncertain thresholds, but identifying accurate thresholds is a biological problem (i.e., determining how much snow wolverines actually need and when), and it shouldn't be addressed by altering the snow modeling process.*

> We agree. "Uncertainties" was removed from this sentence (line 506 – 508).

*Lines 459-460 – This sentence is misleading. Snowfall a few days after May 15 does not "boost wolverine habitat" (see my comment on lines 428-436). The problem that 2015 demonstrates is that single-date thresholds do not accurately depict wolverine denning habitat, not that this particular date is inadequate. Please remove or rephrase.*

> You are correct. Thanks for catching this. This sentence was removed.

*Lines 495-498 – "…which allowed for more gradual changes to wolverine habitat with small changes in SWE" seems to imply that lower sensitivity to changes in thresholds is desired, but sensitivity in itself is not a metric of performance. We should strive for the most accurate, not the least sensitive, snow representation.*

Thanks for pointing this out. We didn't intend to make it sound this way. We removed the text "allowed for" and replaced with "had" (lines 563 – 566).

References:

Fang, Y., Liu, Y., Margulis, S.A., 2020. A New Landsat-era Snow Reanalysis Dataset over the Western United States 2020, C047-0017.

Ma, X., Li, D., Fang, Y., Margulis, S.A., Lettenmaier, D.P., 2023. Estimating spatiotemporally continuous snow water equivalent from intermittent satellite observations: an evaluation using synthetic data. Hydrology and Earth System Sciences 27, 21–38. https://doi.org/10.5194/hess-27-21-2023

Magoun, A.J., Robards, M.D., Packila, M.L., Glass, T.W., 2017. Detecting snow at the den-site scale in wolverine denning habitat. Wildlife Society Bulletin 41, 381–387. https://doi.org/10.1002/wsb.765

Margulis, S.A., Liu, Y., Baldo, E., 2019. A Joint Landsat- and MODIS-Based Reanalysis Approach for Midlatitude Montane Seasonal Snow Characterization. Front. Earth Sci. 7. https://doi.org/10.3389/feart.2019.00272

Pflug, J.M., Margulis, S.A., Lundquist, J.D., 2022. Inferring watershed-scale mean snowfall magnitude and distribution using multidecadal snow reanalysis patterns and snow pillow observations. Hydrological Processes 36, e14581. https://doi.org/10.1002/hyp.14581

Yang, K., Musselman, K.N., Rittger, K., Margulis, S.A., Painter, T.H., Molotch, N.P., 2021. Combining ground-based and remotely sensed snow data in a linear regression model for real-time estimation of snow water equivalent. Advances in Water Resources 104075. https://doi.org/10.1016/j.advwatres.2021.104075

Response to RC2:

We would like to thank anonymous reviewer #2 for their thorough and thoughtful comments. Below, we include detailed responses for each of the reviewer's comments. Line numbers reference where the text changes were made in the manuscript with tracked changes.

*1. Abstract: since the paper builds upon modeled data, I think that the word "model" should be included in this section.*

> This is a good point. In addition to our use of the term "simulations", we made sure to explicitly note that SWE was defined from a reanalysis model (lines 25 – 28). We made sure to define this more frequently in the main text as well.

*2. The authors refer to "annual variations" (e.g., L32, L38, L277, L289, L296, L486, etc.) several times throughout the manuscript, some of which correspond to inter-annual (i.e., year to year) variations, while others correspond to intra-annual (i.e., seasonal) variations. I think this paper would greatly benefit from explicitly stating which type of variability they are referring to.*

> Thanks for pointing this out. We agree that our use of the term "annual" was vague and probably misused at times. We went through the main text and used "annual" to reference differences between years, and "seasonal" was used to reference periods within a year. "Annual" was removed when comparisons included both annual and seasonal timescales.

*3. Figure 1: the map in the top panel could be improved by adding latitude, longitude, north arrow and scale. Since HESS has an international audience, this figure would benefit from adding another map showing the location of Colorado (and RMNP) within the US (the same comment applies to Figure 2).*

> Thanks for the suggestions. We didn't intend for Figure 1 to reference any specific location, but instead show a conceptual of the difference in snow heterogeneity at multiple spatial scales, and the differences in thresholded area between the discretizations when applying a static threshold. Therefore, we don't include any specific coordinates or North references. We also believe that the focus of this figure is to compare the fine-resolution lidar snow patterns, 90m grid cells, and 480m gridcells, which provide scale references. Per your suggestion, we included a North reference in Figure 2, and expanded the reference map to include all US states. The Latitude and Longitude axes in Figure 2b provide the location and scale for this figure.

*4. L134-136: I recommend the authors to show this graphically, at the very least in the Supplementary material.*

> This is a good suggestion. We included this in the revised manuscript's supplementary material (Figure S1).

*5. L197-215: note that previous studies using lidar-derived snow depth have documented the suitability of the normal (He et al. 2019) and gamma (Helbig et al. 2015; Skaugen and Melvold*

*2019; Mendoza et al. 2020) distributions to characterize snow cover in other regions of the world. I think this would be a good place to make this point.*

Good point. We added a passage to this effect at (lines 229 – 232) and included those references, in addition to a 2020 paper from corresponding author Pflug who found normally-distributed snow depth.

6. *L230: I suggest replacing alpha (α) by a different symbol, since this is typically used for snow albedo.*

Thanks for the suggestion. $\alpha$ was replaced with $\beta$ in the manuscript revisions.

7. *L256: what basin are you referring to? Up to this point, no basin description has been provided.*

Thanks for catching this. We didn't focus on average precipitation over any single basin, but instead over the entirety of the study domain. This text was edited in lines 291 – 295.

8. *L262: what do you mean with anomaly? Each climate category?*

That's correct. We like your suggested term "climate category" better. We changed this text, and all following references of "anomalies".

9. *Figure 6: did you try showing these results using symbols connected with straight lines, instead of bars? I think it's hard to visualize differences among configurations. Also, can you please comment on the bimodal behavior shown in Figure 6b? Was this an expected result?*

Thanks for the suggestion. We did try a scatter/line depiction, but found that this approach focused mostly on the slope of the lines between the different years. We tried a number of plotting approaches (e.g. scatter points, whiskers, etc.) but we think that the bar plots better highlight how the thresholded area was most driven by differences in climate, with much smaller variabilities shown by the smaller-magnitude departures between the different discretizations in every year. We agree that this is a lot to look at, so that was the motivation for summarizing the data in subplots b – d.

We didn't expect to see the bimodal distribution in Figure 6b, but was is driven by the larger separations in distributions between warm and cold (6c) and wet and dry (6d) years. We expanded our discussion of this in lines 346 – 359.

10. *Discussion section: I think that the methodological descriptions (e.g., L378-382, L405-417, which are not easy to follow in its current form) and figures presented here should be moved to the Methods and Results sections, respectively.*

Thanks for the suggestion. We moved these sub-methods to Section 3, but colleagues found this text confusing in that area. We think that this is largely because these are secondary tests that were driven by the results that we presented in Section 4. We ended

up moving these passages back into the discussion section, but revised them to make it more clear that we were providing some additional analyses to comprehend the major project results.

*11.   L341-342: it is very difficult to visualize this point from Figure 7. If it is not possible to see this, I suggest adding "not shown" in parentheses.*

Thanks for pointing this out. This passage was edited to clarify that the spike in DF disagreements (which are visible in the difference plots, Figure 7d and Figure 7e) were focused between 2800 and 3200m elevation (lines 389 – 391).

*12.   L381 and L397: how do you quantify the uncertainty in HF? Please clarify.*

Good question. This uncertainty was calculated using the mean absolute differences in DF across the three discretizations. This text edited in lines 431 – 435.

*13.   L465: I think this would be a good place to discuss the main sources of uncertainty in the reanalysis dataset used, referring to the choice of hydrological model structure, parameters and model states, at the very least. Also, it would be good referring the potential impacts that correcting precipitation inputs may have on the SWE dataset and, consequently, on the results and conclusions presented in this paper.*

Thanks for the suggestion. In our revisions to Section 2.2., we included references to model and forcing uncertainties, and discussed how adjustments to the model states, parameters, and forcing (like precipitation) via the reanalysis reweighting have demonstrated the capability to resolve montane snow patterns that are difficult to simulate. However, we stress that the major focus of this manuscript is to investigate how differences in spatial discretizations of snow impact thresholded estimates of habitat support.

14.   L77: replace 'manuscript' by 'paper'.

This was edited (lines 81 – 84).

15.   L94: I suggest replacing "we ask" by "we address the following research questions".

This text was changes as suggested (lines 101 – 105).

16.   LL18: 'included' -> 'include'.

Good catch. This text was changed (lines 130 – 131).